**Data Availability Statement:** This study employed secondary data on COVID-19 hospitalizations from SIVEP-Gripe, a public and open-access database of

# Geographical variation in demand, utilization, and outcomes of hospital services for COVID-19 in Brazil: A descriptive serial cross-sectional study

**Claudia Cristina de Aguiar Pereira[1], Mônica Martins[1], Sheyla Maria Lemos Lima[1], Carla Lourenço Tavares de Andrade[1], Fernando Ramalho Gameleira Soares[1,2], Margareth Crisóstomo Portela**[1] *

**1** Sergio Arouca National School of Public Health, Oswaldo Cruz Foundation (Fiocruz), Rio de Janeiro, RJ, Brazil, **2** Brazilian Institute of Geography and Statistics (IBGE), Rio de Janeiro, RJ, Brazil

* mportela@ensp.fiocruz.br

## Abstract

### Objective

To analyze the geographical variation in the provision of health services, namely in demand, patterns of utilization, and effectiveness in the Brazilian Health Regions in four different periods of the COVID-19 pandemic, from February 2020 to March 2021.

### Methods

Descriptive serial cross-sectional study based on secondary data on COVID-19 hospitalizations from SIVEP-Gripe, a public and open-access database of Severe Acute Respiratory Illness records collected by the Brazilian Ministry of Health, and COVID-19 case notification data from Brasil.io, a repository of public data. Fifty-six epidemiological weeks were split into four periods. The following variables were considered for each Brazilian Health Region, per period: number of hospitalizations, hospitalizations per 100,000 inhabitants, hospitalizations per 100 new cases notified in the Health Region, percentage of hospitalizations with ICU use, percentages of hospitalizations with invasive and non-invasive ventilatory support, percentage of hospitalizations resulting in death and percentage of hospitalizations with ICU use resulting in death. Descriptive statistics of the variables were obtained across all 450 Health Regions in Brazil over the four defined pandemic periods. Maps were generated to capture the spatiotemporal variation and trends during the first year of the COVID-19 pandemic in Brazil.

### Results

There was great variation in how COVID-19 hospitalizations grew and spread among Health Regions, with higher numbers between June and August 2020, and, especially, from mid-December 2020 to March 2021. The variation pattern in the proportion of ICU use in the hospitalizations across the Health Regions was broad, with no intensive care provision in large

Severe Acute Respiratory Illness records collected by the Brazilian Ministry of Health and COVID-19 case notification data from Brasil.io, a repository of public data. The SIVEP-Gripe data for 2020 and 2021 were accessed on the website https://opendatasus.saude.gov.br. The complete daily case notification data for Brazilian during the pandemic were obtained from the website https://brasil.io/home/. All data used in the work are available on https://data.mendeley.com/datasets/2ggr5m5gk4/1.

**Funding:** The authors received no specific funding for this work.

**Competing interests:** The authors have declared that no competing interests exist.

**Abbreviations:** ICU, intensive care unit; SI, Supplemental information; SUS, Unified Health System.

areas in the North, Northeast, and Midwest. The proportions of hospitalizations and hospitalizations with ICU use resulting in deaths were remarkably high, reaching medians of 34.0% and 62.0% across Health Regions, respectively.

## Conclusion

The Heath Regions in Brazil are highly diverse, showing broad disparities in the capacity to respond to the demands imposed by COVID-19, services provided, use and outcomes.

## Introduction

Geographical variation in demand, utilization, and effectiveness of inpatient care is associated with a broad range of individual and contextual factors, among which health needs and service provision preponderate.

In Brazil, there are severe disparities in the provision of health services, in health needs across different segments of the population, and in the access to healthcare, despite the existence of an equitable universal public health system [1]. Social inequities are often expressed in the disease burden, early mortality, and prevalence of multimorbidity that are unfavorable to the less privileged population.

The country has shown one of the worst scenarios on the international stage in the COVID-19 pandemic. Initially, the epidemic caused an increase in cases and deaths geographically circumscribed with distinct temporality, which allowed for some resource sharing and case transfer [2–4]. However, from the end of 2020, the evolution of the pandemic attained national scale, geographically broad and widespread, with transmission rates reaching around 1.20 [5] by March 2021. At that point, a mean of 73,000 new cases and 2,200 deaths were registered daily, and the ICU occupancy rates reached values above 80% and 90%, respectively, in 25 and 17 of the 27 Federative Units (states and Federal District) [6]. The uncontrolled virus circulation had enabled new strains, such as P.1, with estimated transmissibility of 1.4 to 2.2 higher than the original one, contributing to the chaotic pandemic situation in the country [7].

Throughout the evolution of the pandemic in 2021, Brazil held the second place in number of cases, surpassed only by India in April through June of 2021. Moreover, loss of life draws society's attention; there was a rapid and abrupt increase in deaths: over 550 thousand fatalities at the end of July 2021 [8]. In this scenario, the social gradient in the prevalence of COVID-19, access, demand, the use and intensity of services, and the effectiveness of care has been continuously scrutinized [3, 4, 9, 10] and marks the deepening of the historical and perennial social inequality in Brazil. Prospectively, the pandemic acquired unpredictable contours and posed to those already known, other challenges for the organization of care and the network of health services concerning the prevention of contagion, immunization of the population, care of acute cases, recovery and subsequent rehabilitation specific to COVID-19 [11].

Analyses of acute care for cases that required hospitalization by COVID-19 described an uneven pattern in the supply of critical inputs [9, 12], socio-racial inequity [3, 4, 10], and high hospital mortality rates [9, 12]. Regional inequality in the risk of dying during hospitalization indicated a hospital mortality rate in non-elderly patients of 31% in the Northeast region compared to 15% in the South of the country, and still a significant risk of death among intensive care patients and with invasive mechanical ventilation [12]. Specifically, within the Unified Health System (SUS), there was a wide variation of in-hospital mortality associated with

demographic, clinical, and social factors and the structure, quality of health services, and region of the country [9].

The heterogeneity in COVID-19 inpatient care supply and outcomes among macro regions and states has been shown [3, 9, 12, 13], and seems relevant to go beyond, examining differences across Health Regions, which may allow for mitigating distortions that could emerge from looking at municipalities as observation units. A Health Region is made up of a group of municipalities that, in a shared way, organize their resources (such as hospitals, emergency units, and primary care) into care networks for the provision of services to the populations of the municipalities that integrate [13].

The goal of this study is to analyze the geographical variation in the provision of health services, namely in demand, patterns of utilization, and effectiveness, in Brazilian health regions, considering four different periods during the first year of the COVID-19 pandemic in Brazil.

## Methods

### Study design

This is a descriptive serial cross-sectional study based on secondary data (S1 Checklist) [14].

### Setting and participants

This study comprises the COVID-19 hospitalizations in Brazil of patients with symptoms' onset between 23 February 2020 and 20 March 2021. Brazil has about 213 million inhabitants, 26 states, a Federal District, and 5570 municipalities. Its territory is divided into five major geographical regions–North, Northeast, Midwest, Southeast and South–and 450 Health Regions, which are the analysis units in this work.

Considering the epidemiological week (EW) of the case notification, we defined four periods, each including 14 weeks, from EW 09/2020, the official beginning of the pandemic in Brazil, to EW 11/2021. The four periods are the following: period 1, from February 23, 2020 (EW 09/2020) to May 30, 2020 (EW 22/2020); period 2, from May 31, 2020 (EW 23/2020) to September 5, 2020 (EW 36/2020); period 3, from September 6, 2020 (EW 37/2020) to December 12, 2020 (EW 50/2020); and period 4, from December 13, 2020 (EW 51/2020) to March 20, 2021 (EW 11/2021).

### Variables

The following variables were summarized from the data sources described below, per period in each Health Region, considering the hospital's location: number of new confirmed cases, estimated population, number of COVID-19 hospitalizations, number of hospitalizations of patients who live in the Health Region, number of hospitalizations with ICU use, number of hospitalizations with invasive ventilatory support use, number of hospitalizations with non-invasive ventilatory support use, number of inpatient deaths, number of inpatient deaths among ICU users. Subsequently, other variables were calculated. Three variables were included to identify the hospitalizations' occurrence: COVID-19 hospitalizations per 100,000 inhabitants, rate of hospitalizations per 100 new cases notified in the Health Region–it is not a proportion, once patients from other areas may also be hospitalized–, and percentage of COVID-19 hospitalizations of patients who live in the Health Region. Three other variables were built as healthcare process proxies: percentage of hospitalizations with ICU use, percentage of hospitalizations with invasive ventilatory support, and percentage of hospitalizations with non-invasive ventilatory support. Moreover, two variables were calculated to account for inpatient care outcomes: percentage COVID-19 hospitalizations resulting in death for

COVID-19 and percentage of COVID-19 hospitalizations with ICU use resulting in death for COVID-19.

## Data sources

Data were extracted from two main sources: the SIVEP-Gripe, a public and open-access database, comprised of prospectively collected data on Severe Acute Respiratory Illness records and maintained by the Brazilian Ministry of Health; and COVID-19 case notification data from Brasil.io, a repository of public data. The SIVEP-Gripe data for 2020 and 2021 were accessed on the website https://opendatasus.saude.gov.br, on April 19, 2021. The limit of the study's observation period was set accounting for the availability of more reliable data with a delay of 4–5 weeks because of the records' flow. The complete daily case notification data were obtained from the website https://brasil.io/home/ on April 24, 2021.

Observations from the SIVEP-Gripe database were filtered based on two variables: an indicator of hospitalization occurrence and the final case classification as 'SRAG by COVID-19'.

Both datasets include variables that indicate the EW in which the case was notified. From SIVEP-Gripe, we also extracted the codes of the hospitals' and the patients' municipalities, and variables indicating ICU use, ventilatory support use (invasive or non-invasive), and hospitalization outcome, including death for COVID-19. From the COVID-19 case notification data, we obtained the variables indicating the number of new confirmed cases, the municipality's code and estimated population. Despite the availability of the Health Regions' codes for the patients' and hospitals' municipalities in the SIVEP-Gripe, we observed some missing values and opted for using an additional dataset associating municipalities and Health Regions from DATASUS [IT Department of the Brazilian Unified Health System].

## Data analysis

Descriptive statistics of the variables listed above were obtained across the 450 Health Regions in Brazil over the four defined pandemic periods. Box plots were employed to allow for visualization of any changes in their magnitude and variability over time. Furthermore, maps were used to capture the trends in spatiotemporal variables during the first year of the COVID-19 pandemic in Brazil. Data management and analyses were performed using SAS® statistical package, and all data and SAS program are publicly available [15].

## Results

The descriptive statistics of the variables selected show the broad variation among the 450 Health Regions in the country (Table 1). Results regarding the rate of COVID-19 hospitalizations per 100 new cases in the Health Region in the first period accounted only for 445 regions because five did not report any cases. Moreover, the variables defined in terms of the COVID-19 hospitalizations' proportions were not computed when there were no hospitalizations, implying the exclusion of 29, 7, 14, and 8 regions in the first, second, third, and fourth period, respectively. Two regions in Northern Brazil, in the states of Acre and Roraima, and two others in Northeastern Brazil, in the state of Piaui, had no hospitalizations in any period, despite the occurrence of a relevant number of cases.

Table 1 and Figs 1 and 2 indicate the growth and spread of COVID-19 hospitalizations from period 1 to 2, fall in period 3, and numbers were skyrocketing in almost the entire country in period 4.

The interquartile range for 'hospitalizations per 100,000 inhabitants' varied between 3.2 and 19.5, in period 1, between 31.4 and 110.6, in period 2, between 20.9 and 95.5, in period 3, and between 38.7 and 184.1, in period 4. It ratifies the high variation among Health Regions

**Table 1. Distribution of variables related to the occurrence of COVID-19 hospitalizations and inpatient care process and outcomes in the administrative Health Regions.** Brazil, epidemiological weeks 09/2020–11/2021, 23 Feb 2020–20 Mar 2021.

| Variable | Period | N | Mean | Std | Min | Q1 | Median | Q3 | Max |
|---|---|---|---|---|---|---|---|---|---|
| COVID-19 hospitalizations | 1 | 450 | 276.0 | 1,659.5 | 0 | 6 | 18.0 | 79 | 26,635 |
| | 2 | 450 | 578.3 | 1,787.5 | 0 | 54 | 150.5 | 370 | 28,764 |
| | 3 | 450 | 397.4 | 1,202.9 | 0 | 36 | 124.0 | 305 | 16,800 |
| | 4 | 450 | 665.2 | 1,785.5 | 0 | 69 | 224.5 | 596 | 28,073 |
| Rate of COVID-19 hospitalizations per 100,000 inhabitants | 1 | 450 | 20.4 | 34.1 | 0.0 | 3.3 | 8.1 | 19.6 | 271.3 |
| | 2 | 450 | 80.9 | 66.2 | 0.0 | 31.4 | 66.8 | 110.6 | 399.2 |
| | 3 | 450 | 62.7 | 52.5 | 0.0 | 20.9 | 52.5 | 95.5 | 294.7 |
| | 4 | 450 | 116.1 | 94.4 | 0.0 | 38.7 | 95.7 | 184.1 | 498.6 |
| Rate of hospitalizations per 100 new cases of COVID-19 | 1 | 445 | 17.5 | 15.4 | 0.0 | 6.2 | 14.0 | 25.7 | 129.9 |
| | 2 | 450 | 5.7 | 4.1 | 0.0 | 2.5 | 5.2 | 8.1 | 23.5 |
| | 3 | 450 | 4.8 | 3.6 | 0.0 | 1.9 | 4.3 | 7.1 | 22.7 |
| | 4 | 450 | 4.8 | 3.5 | 0.0 | 2.1 | 4.5 | 6.9 | 22.6 |
| Proportion (%) of hospitalizations of residents in the health area | 1 | 421 | 72.8 | 23.1 | 0.0 | 59.8 | 77.3 | 91.4 | 100.0 |
| | 2 | 443 | 72.0 | 18.8 | 0.0 | 60.9 | 74.1 | 85.8 | 100.0 |
| | 3 | 436 | 69.1 | 20.4 | 3.4 | 55.7 | 72.1 | 84.5 | 100.0 |
| | 4 | 442 | 68.9 | 19.9 | 4.3 | 57.8 | 71.0 | 83.2 | 100.0 |
| Proportion (%) of hospitalizations with ICU use | 1 | 421 | 27.7 | 22.1 | 0.0 | 8.6 | 28.2 | 41.2 | 100.0 |
| | 2 | 443 | 27.6 | 19.4 | 0.0 | 11.6 | 26.8 | 39.5 | 96.3 |
| | 3 | 436 | 31.0 | 21.1 | 0.0 | 16.6 | 30.4 | 42.3 | 100.0 |
| | 4 | 442 | 29.1 | 20.7 | 0.0 | 13.9 | 28.0 | 39.6 | 92.2 |
| Proportion (%) of hospitalizations with invasive ventilatory support use | 1 | 421 | 20.1 | 18.3 | 0.0 | 7.7 | 16.7 | 27.8 | 100.0 |
| | 2 | 443 | 18.1 | 13.3 | 0.0 | 9.9 | 16.1 | 23.7 | 85.9 |
| | 3 | 436 | 16.9 | 13.0 | 0.0 | 8.4 | 14.8 | 22.3 | 93.7 |
| | 4 | 442 | 18.8 | 14.1 | 0.0 | 10.0 | 16.0 | 24.7 | 100.0 |
| Proportion (%) of hospitalizations with non-invasive ventilatory support use | 1 | 421 | 38.2 | 22.7 | 0.0 | 23.1 | 37.5 | 50.0 | 100.0 |
| | 2 | 443 | 44.9 | 17.6 | 0.0 | 35.0 | 46.5 | 55.8 | 100.0 |
| | 3 | 436 | 49.4 | 20.3 | 0.0 | 38.2 | 51.3 | 62.6 | 100.0 |
| | 4 | 442 | 52.1 | 20.0 | 0.0 | 41.0 | 55.6 | 65.6 | 100.0 |
| Proportion (%) of COVID-19 hospitalizations resulting in death | 1 | 421 | 38.3 | 24.1 | 0.0 | 21.1 | 34.1 | 50.0 | 100.0 |
| | 2 | 443 | 36.2 | 17.3 | 0.0 | 24.9 | 32.6 | 42.9 | 100.0 |
| | 3 | 436 | 32.4 | 18.8 | 0.0 | 20.8 | 28.6 | 37.6 | 100.0 |
| | 4 | 442 | 38.0 | 19.2 | 0.0 | 27.5 | 34.0 | 43.9 | 100.0 |
| Proportion (%) of COVID-19 hospitalizations with ICU use resulting in death | 1 | 345 | 59.3 | 29.0 | 0.0 | 40.0 | 57.1 | 81.8 | 100.0 |
| | 2 | 403 | 62.2 | 19.6 | 0.0 | 50.0 | 60.8 | 74.4 | 100.0 |
| | 3 | 391 | 55.2 | 21.2 | 0.0 | 42.4 | 53.1 | 66.7 | 100.0 |
| | 4 | 406 | 62.3 | 21.7 | 0.0 | 49.4 | 62.0 | 76.5 | 100.0 |

Sources: SIVEP-Gripe, Brazilian Minister of Health, and Brasil.io.

Periods: 1, epidemiological weeks 09-22/2020 (23 Feb 2020–30 May 2020); 2, epidemiological weeks 23-36/2020 (31 May 2020–05 Sep 2020); 3, epidemiological weeks 37-50/2020 (06 Sep 2020–12 Dec 2021); and 4, epidemiological weeks 51/2020-11/2021 (13 Dec 2021–20 Mar 2021).

and, overall, more inpatient care utilization for COVID-19 in period 2, and, especially, in period 4 (Fig 2). With regard to the rate of hospitalizations per 100 new cases of COVID-19 in the Health Regions, there was a reduction in its variability throughout the periods analyzed. The interquartile range was 6.2 to 25.7 in the first period and 2.1 to 6.9 in the fourth period (Table 1, Fig 2).

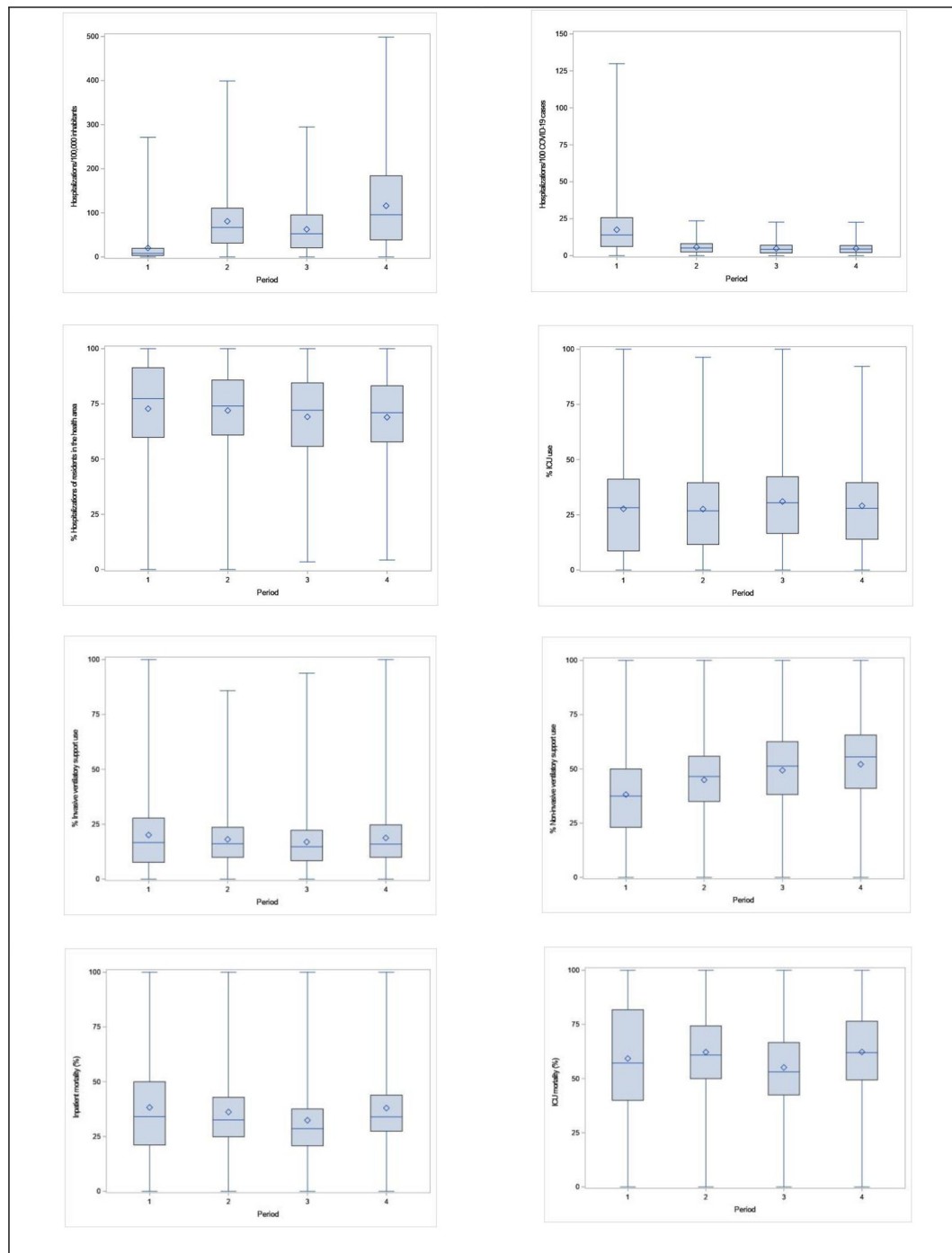

**Fig 1. Box-plots of variables related to the occurrence of COVID-19 hospitalizations and inpatient care process and outcomes in the administrative Health Regions.** Brazil, epidemiological weeks 09/2020–11/2021, 23 Feb 2020–20 Mar 2021. Sources: SIVEP-Gripe, Minister of Health, Brazil; and Brasil.io. From left to right, top to bottom: COVID-19 hospitalizations per 100,000 inhabitants; COVID-19 hospitalizations per 100 new COVID-19 cases; Proportion of hospitalizations of residents in the Health Area; Proportion of hospitalizations with ICU use; Proportion of hospitalizations with invasive ventilatory support; Proportion of hospitalizations with non-invasive ventilatory support; Proportion of hospitalizations resulting in death; Proportion of hospitalizations with ICU use resulting in death. Periods: 1, epidemiological weeks 09-22/2020 (23 Feb 2020–30 May 2020); 2, epidemiological weeks 23-36/2020 (31 May 2020–05 Sep 2020); 3, epidemiological weeks 37-50/2020 (06 Sep 2020–12 Dec 2021); and 4, epidemiological weeks 51/2020–11/2021 (13 Dec 2021–20 Mar 2021).

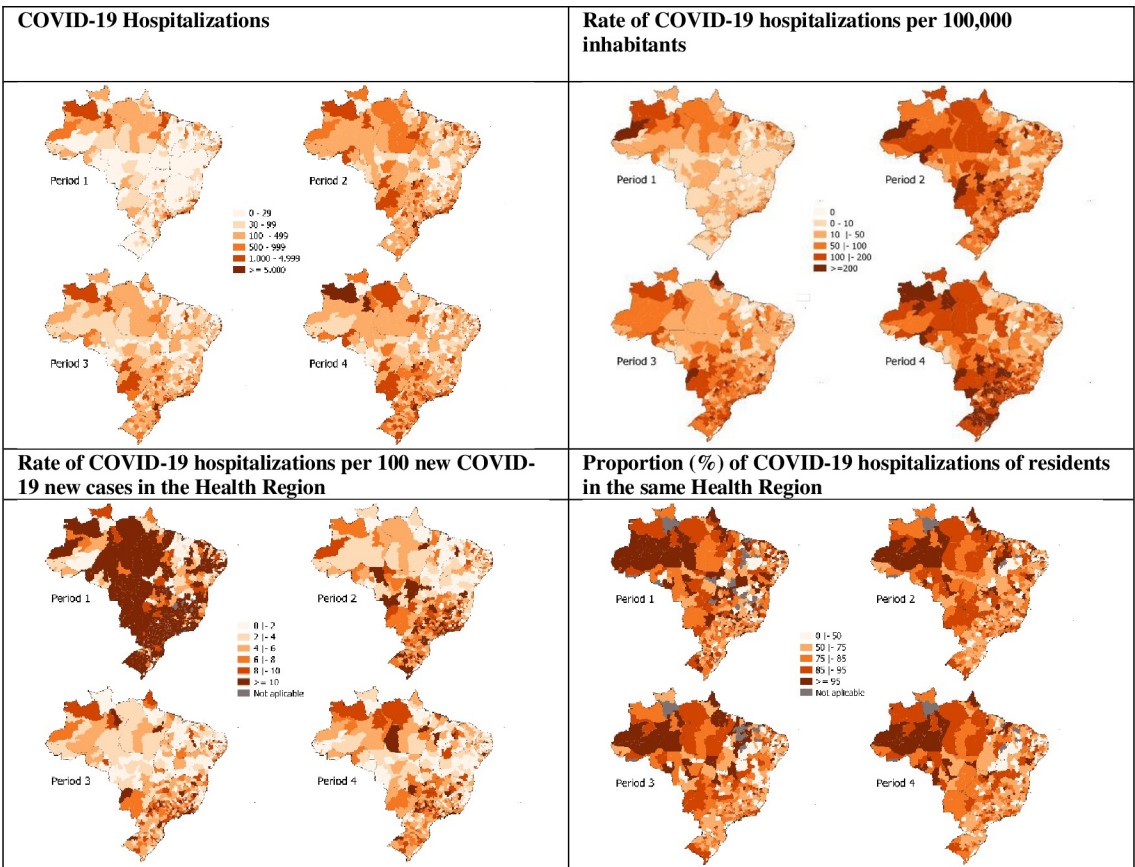

**Fig 2. Spatiotemporal occurrence of COVID-19 hospitalizations.** Brazil, epidemiological weeks 09/2020–11/2021, 23 Feb 2020–20 Mar 2021. Sources: SIVEP-Gripe, Minister of Health, Brazil; and Brasil.io. Periods: 1, epidemiological weeks 09-22/2020 (23 Feb 2020–30 May 2020); 2, epidemiological weeks 23-36/2020 (31 May 2020–05 Sep 2020); 3, epidemiological weeks 37-50/2020 (06 Sep 2020–12 Dec 2021); and 4, epidemiological weeks 51/2020–11/2021 (13 Dec 2021–20 Mar 2021).

The proportion of COVID-19 hospitalizations in the Health Regions of residents of their own decreased, over the periods, whereas attendance of non-residents increased. The median of the variable among the Health Regions decreased from 77.3% to 71.0%. Higher proportions of residents were predominantly observed among hospitalizations in the Health Regions of Northern and Midwestern Brazil, especially in the fourth period.

The median and the interquartile ranges of the variable 'ICU use' in the Health Regions, in the four periods, are also shown in Table 1. The pattern of the variation in the proportion of ICU use in the hospitalizations across the Health Regions (Fig 1) was broad, especially in the first period, when the states and municipalities of the country were organizing their capacity to attend to the demands brought by the COVID-19 pandemic, including the supply of ICU beds. The highest median in the third period, marked by the reduction of hospitalizations, suggests the possibility of expanding ICU use in the scenario of lower competition for complex care. Complementarily, maps in Fig 3 show that intensive care was not provided in large areas of the country, especially in the North, Northeast, and Midwest. Two other healthcare process variables were examined, and the results suggest a slight variation in invasive ventilatory support use and consistent growth in non-invasive ventilatory support use over time (Table 1, Fig 3).

Finally, the proportions of COVID-19 hospitalizations and COVID-19 hospitalizations with ICU use resulting in death indicate remarkably high levels of inpatient mortality (Fig 4).

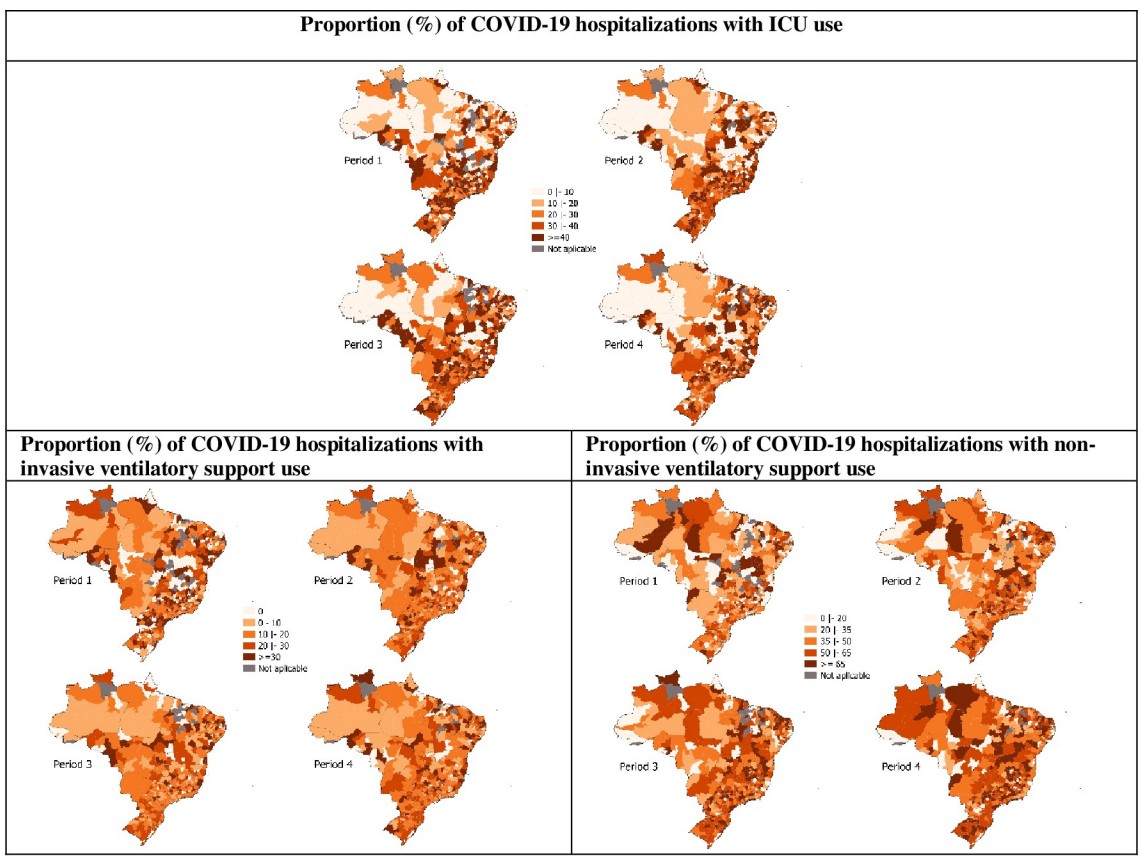

**Fig 3. Spatiotemporal distribution of ICU, invasive, and non-invasive ventilatory support use among COVID-19 hospitalizations.** Brazil, epidemiological weeks 09/2020–11/2021, 23 Feb 2020–20 Mar 2021. Sources: SIVEP-Gripe, Minister of Health, Brazil. Periods: 1, epidemiological weeks 09-22/2020 (23 Feb 2020–30 May 2020); 2, epidemiological weeks 23-36/2020 (31 May 2020–05 Sep 2020); 3, epidemiological weeks 37-50/2020 (06 Sep 2020–12 Dec 2021); and 4, epidemiological weeks 51/2020–11/2021 (13 Dec 2021–20 Mar 2021).

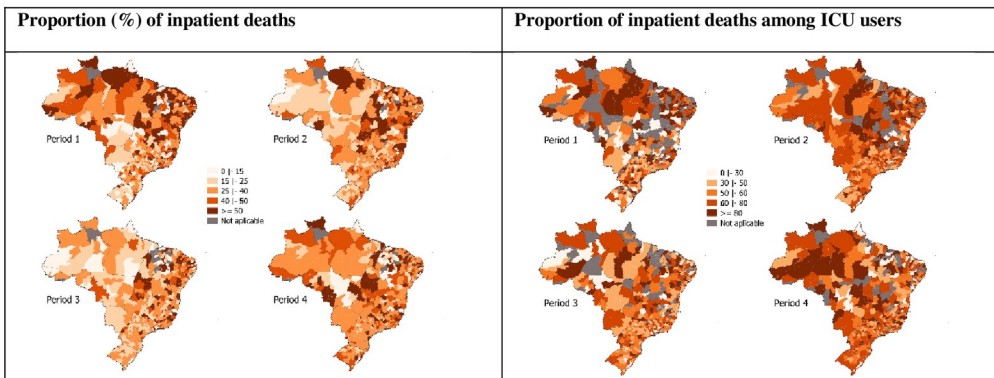

**Fig 4. Spatiotemporal distribution of COVID-19 inpatient general and ICU mortality.** Brazil, epidemiological weeks 09/2020–11/2021, 2020–11/2021, 23 Feb 2020–20 Mar 2021. Sources: SIVEP-Gripe, Minister of Health, Brazil. Periods: 1, epidemiological weeks 09-22/2020 (23 Feb 2020–30 May 2020); 2, epidemiological weeks 23-36/2020 (31 May 2020–05 Sep 2020); 3, epidemiological weeks 37-50/2020 (06 Sep 2020–12 Dec 2021); and 4, epidemiological weeks 51/2020–11/2021 (13 Dec 2021–20 Mar 2021).

In both cases, there was a reduction in the variation of the indicators from the first period to the subsequent ones. The median of the proportions of COVID-19 hospitalizations resulting in deaths in the Health Regions fell from period 1 to period 2 (34.1% to 32.6%) and from period 2 to period 3 (28.6%). From period 3 to 4 (34.0%), it returned to the level of period 1. The median of the other indicator, in turn, increased from period 1 (57.1%) to period 2 (60.8%), decreased from period 2 to period 3 (53.1%), and increased again, reaching its highest value from period 3 to 4 (62.0%).

## Discussion

The drawn picture reflects COVID-19 spreading and stresses inequalities in the access to healthcare in the Brazilian pandemic context. Geographically, demand, and access conformed use patterns and effectiveness that were regionally and socially disparate. Brazil has about 6000 hospitals, but the majority with less than 50 beds and very low complexity profile. Beds, intensive care, equipment, supplies, and trained professional teams are highly concentrated in metropolitan areas, especially in the richest cities. Just before the pandemic, 90.4% of the municipalities and 27.6% of the Health Regions, mainly in the North and in the Northeast, had no ICU beds [13, 16]. The findings for these regions evidenced the incapacity of the country, in the absence of a national coordination, to mitigate historically known and worrisome disparities, which acquired sharpness and escalated in the COVID-19 pandemic scenario [13, 16, 17]. COVID-19 has disproportionately affected states and municipalities with higher socioeconomic vulnerabilities [4], and, in large urban centers, the areas and population groups socially vulnerable [18, 19].

The massive difference between the mean and the median of the number of COVID-19 hospitalizations denotes their concentration in some few regions, such as the metropolitan areas of São Paulo [20] and Rio de Janeiro, whose big numbers push up the mean, while many other regions show much lower numbers. Over time, a high volume of hospitalizations occurred in highly populated metropolitan areas, some of them with an elevated provision of healthcare resources, which contrast with many other geographically extensive regions, with low populational densities and low healthcare resources, in the interior of the country, especially in the North, Northeast, and Midwest. These regions often reported small numbers of hospitalizations.

The rates of hospitalizations per 100,000 inhabitants and hospitalizations per 100 new cases of COVID-19 in the Health Regions provide complementary views of the intensity in which hospitalizations occurred. In the first case, elevated values may be related to high volumes of hospitalizations, even relatively to large populations such as those of the metropolitan areas. Such areas that were strongly affected by the pandemic were also references for the provision of healthcare. Elevated values of the indicator may also be related to Health Regions in low population density areas searched for relatively better healthcare structures.

The rate of hospitalizations per 100 new cases registered in the Health Region, in turn, despite not being conceived as a proportion, provides a proxy for the extent that COVID-19 cases result in hospitalization. Extremely low values may reflect underuse of inpatient care, while high values are likely to reflect either low availability of tests or provision of inpatient care to residents of other Health Regions. The number of new cases in Brazil is likely to be highly underestimated. As Kameda and collaborators [21] pointed out, the national testing approach has not been adequate in the country, primarily due to a shortage of tests and reagents caused by a lack of coordination and anticipation of reagent purchasing. Also, there has been fragmented funding and distribution of tests throughout the country. Nevertheless, since the second semester of 2020, some initiatives were established to expand the testing capacity at SUS.

In 50% of the Health Regions, the proportion of hospitalizations of non-residents was at least 23–29%, over the periods. This suggests a high number of Brazilians has needed to search for COVID-19 hospital care in other Health Regions. In a country with continental dimensions, the need to travel long distances in search of care and hospitalization, in other municipalities within the same Health Region or not, certainly generated a detrimental situation for COVID-19 patients, given the characteristics of the disease and how it can rapidly evolve [9, 22].

The increase in the proportion of hospitalizations with ICU use and reduction of hospitalizations resulting in death in the third period suggest that the remarkably high demand brought by critical moments may itself have contributed to a reduction in the patients' access to needed healthcare, and to worse outcomes. The fourth and last period of the analysis was the most severe one for most indicators analyzed. Hospitalizations resulting in death in the fourth period were slightly lower in the fourth period than in the first one. The fourth period coincides with the holiday season and months after that, when new strains started to circulate, including the P.1 lineage, which has drastically affected the population, including younger age groups [7]. The proportion of hospitalizations with non-invasive ventilatory support use steadily increased from the first to the fourth period (38.2% to 52%), whereas invasive ventilatory support did not present a clear pattern in the periods considered. The increasing use of non-invasive ventilatory support may be related to hospitalized patients' morbidity, age profile, and COVID-19 care learning.

The mean proportion of COVID-19 hospitalizations resulting in death, across Health Regions, ranged from 32.4% in the third period to 38.3% in the first period. Although these results considering the Health Regions as units of analysis are not directly comparable with those focusing on the hospitalizations as units of analyses, it is interesting to note that they are not divergent. Another study, also using SIVEP-Gripe data and considering the period from February to August of 2020, found a total in-hospital mortality of 38% in Brazil [10], with a variation from 31% in the South to 50% in the North [10, 12]. The mean proportion of COVID-19 hospitalizations with ICU use resulting in death in our study varied from 55.2% in the third period to 62.3% in the fourth period. The study carried out by Ranzani and colleagues found that the ICU mortality in the country was 55%, ranging from 53% in the South to 79% in the North [12]. Using the Hospital Information System (SIH) of the Unified Health System (SUS), another work showed the proportion of deaths among those who used the ICU was 55.7% [9].

Variability in hospital mortality rates between Brazilian states was observed in other studies. Explanations include the insufficiency of health resources, especially for complex healthcare [13, 16], excessive burden on the health service network, and insufficient regional/local capacity to coordinate actions to deal with COVID-19, in the absence of national coordination able to mitigate the major regional differences in an immense and diverse country [4, 9, 12].

Although our analyses considered Health Regions comprised of a set of municipalities, it is necessary to emphasize the small number of hospitalizations in some of them. Such small numbers affected the statistics of the variables that had them as denominators. For instance, many observations in which the percentage of inpatient deaths reached 100% were related to those cases.

The results from our analyses should be looked at with caution as they do not provide estimates for the country or the macro-regions. It is fundamental to have in mind that Health Regions are highly diverse, including territory size, number of municipalities they serve, socioeconomic profile and availability of health resources.

Besides, this study has limitation due to the study design, mainly for the observational, descriptive, and database nature. Although SIVEP-Gripe data, being the only data aggregating

all COVID-19 hospitalizations in Brazil, have been highly relevant to delineate COVID-19 course in the country, they have some reliability problems. ICU use, for example, may be over-estimated in some Health Regions with scarcity of ICU beds, which sometimes register the use of intermediary care units' structures as ICU use. There is also loss of data regarding the hospitalization's outcome. Additionally, the descriptive nature of the study does not allow for controlling of confounding and specific bias. Thereby, further studies with longer observation periods are required to understand the impact of COVID-19 on the social, health and health care inequalities.

Nevertheless, this work provides a good picture of COVID-19 spreading and hospitalizations in the first year of the pandemic in Brazil. The end of the observation period coincided with the worst moment of the pandemic in the country, marked by the collapse of the healthcare system [6]. The following two months, however, were still very critical, and should be further examined. The vaccination, that was initiated on January 18[th] with focus on the healthcare professionals and more vulnerable populations groups (the elderly and those with comorbidities), has been expanded. Since June 2021, epidemiological indicators and ICU occupancy rates have consistently improved in the country. As in other countries, there are still concerns about variants such as Delta, and the loss of vaccines' effectiveness over time among the elderly, but the scenario is unquestionably better.

The COVID-19 pandemic acquired unpredictable contours in Brazil, with the resulting sanitary and humanitarian crisis poorly controlled in a hard and controversial political context. COVID-19 added, to those already known, other challenges for the organization of care and the network of health services concerning the prevention of contagion, care of acute cases, recovery and subsequent rehabilitation needs, and immunization of the population. Questions are still continuously posed, and extra challenges to healthcare organization arise from the need to attend the repressed demand during the pandemic's period, and the demand brought by Long Covid.

## Supporting information

**S1 Checklist. STROBE checklist.**
(DOC)

## Acknowledgments

MCP, CCAP, and MM are recipients of productivity fellowships from the Brazilian National Council of Scientific and Technological Development (CNPq).

## Author Contributions

**Conceptualization:** Claudia Cristina de Aguiar Pereira, Mônica Martins, Sheyla Maria Lemos Lima, Carla Lourenço Tavares de Andrade, Margareth Crisóstomo Portela.

**Formal analysis:** Claudia Cristina de Aguiar Pereira, Mônica Martins, Sheyla Maria Lemos Lima, Carla Lourenço Tavares de Andrade, Fernando Ramalho Gameleira Soares, Margareth Crisóstomo Portela.

**Methodology:** Claudia Cristina de Aguiar Pereira, Mônica Martins, Sheyla Maria Lemos Lima, Carla Lourenço Tavares de Andrade, Margareth Crisóstomo Portela.

**Writing – original draft:** Claudia Cristina de Aguiar Pereira, Mônica Martins, Sheyla Maria Lemos Lima, Carla Lourenço Tavares de Andrade, Fernando Ramalho Gameleira Soares, Margareth Crisóstomo Portela.

**Writing – review & editing:** Claudia Cristina de Aguiar Pereira, Mônica Martins, Sheyla Maria Lemos Lima, Carla Lourenço Tavares de Andrade, Margareth Crisóstomo Portela.

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
