## [Decision Letter · Decision Letter 0]

28 Jul 2021

PONE-D-21-15377

Geographical variation in demand, utilization, and outcomes of hospital services: COVID-19, Brazil

PLOS ONE

Dear Dr. Portela,

Thank you for submitting your manuscript to PLOS ONE. After careful consideration, we feel that it has merit but does not fully meet PLOS ONE’s publication criteria as it currently stands. Therefore, we invite you to submit a revised version of the manuscript that addresses the points raised during the review process.

This is quite an interesting investigation, exploring health inequities in Brazil. The results are well described. Reasons for thee inequities should be discussed and/or hypothesized. The Authors should also discuss what were the change across thee periods. A paragraph dealing with future lines of health utilization should be added 

A rebuttal letter that responds to each point raised by the academic editor and reviewer(s). You should upload this letter as a separate file labeled 'Response to Reviewers'.A marked-up copy of your manuscript that highlights changes made to the original version. You should upload this as a separate file labeled 'Revised Manuscript with Track Changes'.An unmarked version of your revised paper without tracked changes. You should upload this as a separate file labeled 'Manuscript'.If applicable, we recommend that you deposit your laboratory protocols in protocols.io to enhance the reproducibility of your results. Protocols.io assigns your protocol its own identifier (DOI) so that it can be cited independently in the future. For instructions see: http://journals.plos.org/plosone/s/submission-guidelines#loc-laboratory-protocols. Additionally, PLOS ONE offers an option for publishing peer-reviewed Lab Protocol articles, which describe protocols hosted on protocols.io. Read more information on sharing protocols at https://plos.org/protocols?utm_medium=editorial-email&utm_source=authorletters&utm_campaign=protocols.

We look forward to receiving your revised manuscript.

Kind regards,

Chiara Lazzeri

Academic Editor

PLOS ONE

2. Please note that in order to use the direct billing option the corresponding author must be affiliated with the chosen institute. Please either amend your manuscript to change the affiliation or corresponding author, or email us at plosone@plos.org with a request to remove this option.

3. We note that Figures2,3 & 4in your submission contain [map/satellite] images which may be copyrighted. All PLOS content is published under the Creative Commons Attribution License (CC BY 4.0), which means that the manuscript, images, and Supporting Information files will be freely available online, and any third party is permitted to access, download, copy, distribute, and use these materials in any way, even commercially, with proper attribution. For these reasons, we cannot publish previously copyrighted maps or satellite images created using proprietary data, such as Google software (Google Maps, Street View, and Earth). For more information, see our copyright guidelines: http://journals.plos.org/plosone/s/licenses-and-copyright.

a. You may seek permission from the original copyright holder of Figures 2,3 & 4 to publish the content specifically under the CC BY 4.0 license. 

Additional Editor Comments (if provided):

Reviewers' comments:

Reviewer's Responses to Questions

**Comments to the Author**

1. Is the manuscript technically sound, and do the data support the conclusions?

Reviewer #1: Yes

2. Has the statistical analysis been performed appropriately and rigorously? 

Reviewer #1: Yes

3. Have the authors made all data underlying the findings in their manuscript fully available?

Reviewer #1: No

4. Is the manuscript presented in an intelligible fashion and written in standard English?

Reviewer #1: No

5. Review Comments to the Author

Reviewer #1: Dear Collegues, thank you very much for the opportunity to review your scientific paper which tries to elucidate the geographical differences that exist in Brazil in relation to the COVID-19 pandemic.

However the paper is not written following strobe guidelines, therefore it is very difficult to understand and read the methodology and the differences given in the scientific paper.

When the changes are made, the paper can be reviewed again without any problem.

6. PLOS authors have the option to publish the peer review history of their article (what does this mean?). If published, this will include your full peer review and any attached files.

Reviewer #1: No

---

## [Author Response · Author response to Decision Letter 0]

27 Aug 2021

Dear Dr. Chiara Lazerri,

Thank you for your feedback regarding our manuscript submitted to PLOS ONE. We appreciated the careful considerations and comments to improve it. All points were addressed, and some parts of the text were reorganized in order to make them clearer and concise. Therefore, we are submitting a revised version accounting for the comments and reflections they raised. 

In taking into account STROBE guidelines, we proposed a change in the title of the manuscript (Geographical variation in demand, utilization, and outcomes of hospital services for COVID-19 in Brazil: a descriptive serial cross-sectional study), made the decision of reorganizing the Methods, and added some relevant information. All the data used in the work were deposited in a public repository. We transferred some paragraphs from the Introduction and the Results to Discussion, making them more concise. With regard to the Discussion, we reorganized the text, excluding parts that were not adding relevant information, making clearer interpretations of the findings and the reasons for the inequities, and adding considerations about what happened after the observation period. 

Kind regards,

Margareth Crisóstomo Portela, PhD

Corresponding Author

 

Responses regarding journal requirements:

Done.

2. Please note that in order to use the direct billing option the corresponding author must be affiliated with the chosen institute. Please either amend your manuscript to change the affiliation or corresponding author, or email us at plosone@plos.org with a request to remove this option.

OK.

3. We note that Figures2,3 & 4in your submission contain [map/satellite] images which may be copyrighted. All PLOS content is published under the Creative Commons Attribution License (CC BY 4.0), which means that the manuscript, images, and Supporting Information files will be freely available online, and any third party is permitted to access, download, copy, distribute, and use these materials in any way, even commercially, with proper attribution. For these reasons, we cannot publish previously copyrighted maps or satellite images created using proprietary data, such as Google software (Google Maps, Street View, and Earth). For more information, see our copyright guidelines: http://journals.plos.org/plosone/s/licenses-and-copyright.

WE WOULD LIKE TO CLARIFY THAT ALL IMAGES (MAPS) PRESENTED IN OUR PAPER WERE CREATED BY US, FROM OUR ANALYSIS OF THE DATA DESCRIBED IN METHODS. THEREFORE, THEY ARE FULLY OURS AND, UPON ACCEPTANCE OF THE MANUSCRIPT, THEY CAN BE PUBLISHED UNDER CC BY 4.0. 

a. You may seek permission from the original copyright holder of Figures 2,3 & 4 to publish the content specifically under the CC BY 4.0 license. 

Maps at the CIA (public domain):https://www.cia.gov/library/publications/the-world-factbook/index.html and https://www.cia.gov/library/publications/cia-maps-publications/index.html

NASA Earth Observatory (public domain):http://earthobservatory.nasa.gov/

Considering the answer provided in item 3, the points above are not applicable.

Comments to the Author

1. Is the manuscript technically sound, and do the data support the conclusions?

Reviewer #1:Yes________________________________________

2. Has the statistical analysis been performed appropriately and rigorously?

Reviewer #1: Yes________________________________________

3. Have the authors made all data underlying the findings in their manuscript fully available?

Reviewer #1: No

Our response: All the data used in the work were deposited in a public repository https://data.mendeley.com/datasets/2ggr5m5gk4/1.________________________________________

5. Is the manuscript presented in an intelligible fashion and written in standard English?

Reviewer #1:No

Our response: We have revised the manuscript accordingly.________________________________________

5. Review Comments to the Author

Reviewer #1: Dear Colleagues, thank you very much for the opportunity to review your scientific paper which tries to elucidate the geographical differences that exist in Brazil in relation to the COVID-19 pandemic.

However, the paper is not written following strobe guidelines, therefore it is very difficult to understand and read the methodology and the differences given in the scientific paper.

When the changes are made, the paper can be reviewed again without any problem.

Our response: Thank you for your availability to review our manuscript. We have added the STROBE checklist as a supplemental information, indicating where each item is found in the manuscript. We reorganized the Methods in order to comply more strictly with the guidelines. A few items were not applicable to the type of study proposed. Additionally, we would like to underline that we classify our study as a health services research manuscript, providing a thorough description of the background motivation for the work, our methods and results.________________________________________

6. PLOS authors have the option to publish the peer review history of their article (what does this mean?). If published, this will include your full peer review and any attached files.

Do you want your identity to be public for this peer review? For information about this choice, including consent withdrawal, please see our Privacy Policy.

Reviewer #1: No

Done.

---

## [Editor Report · Decision Letter 1]

7 Sep 2021

Geographical variation in demand, utilization, and outcomes of hospital services for COVID-19 in Brazil: a descriptive serial cross-sectional study

PONE-D-21-15377R1

Dear Dr. Portela,

We’re pleased to inform you that your manuscript has been judged scientifically suitable for publication and will be formally accepted for publication once it meets all outstanding technical requirements.

Kind regards,

Chiara Lazzeri

Academic Editor

PLOS ONE
---

## [Editor Report · Acceptance letter]

22 Sep 2021

PONE-D-21-15377R1 

Geographical variation in demand, utilization, and outcomes of hospital services for COVID-19 in Brazil: a descriptive serial cross-sectional study 

Dear Dr. Portela:

I'm pleased to inform you that your manuscript has been deemed suitable for publication in PLOS ONE. Congratulations! Your manuscript is now with our production department. 

Kind regards, 

on behalf of

Dr. Chiara Lazzeri 

Academic Editor

PLOS ONE